# OpenReview forum: "Position: Anthropomorphic Misalignment Research Needs Stronger Evidence"
_ICML.cc/2026/Position_Paper_Track — ICML 2026 Position Paper Track spotlight_

### Official Review · Reviewer_uKr5 · 2026-02-23

**Significance:** 3
**Argument Clarity:** 2
**Ethics Flag:** Yes
**Rating:** 4
**Confidence:** 4

**Questions:**

Can you provide comments on the following:

* Clarify operationally what would count as convincing L3 evidence in transformers (minimum intervention/robustness bar, examples of successful L3).
* Provide a concrete framing for when L1/L2 evidence is adequate for deployment/regulatory recommendations.

Also please refer to some of my questions earlier. The main problem I find for this paper is: I like the opinion but not sure if this paper add anything for the community other than raising the problem statement again.

**Alternative Views Section:**

Yes

**Compliance With Llm Reviewing Policy A Conservative:**

Affirmed.

**Discussion Potential:**

2

**Ethical Review Concerns:**

the content should be checked for ethical reviews due to relevant to AI safety and policy

**Ethics Review Area:**

["Other Expertise"]

**Final Justification:**

While I am still not fully convinced by the current level in terms of contribution, I appreciate the effort to make interpretability research more rigorous and scientific and I am willing to support this, and I see this paper as a step in that direction. Hence, I will increase my score to 4

**Paper Summary:**

In this paper, the authors investigate the general domain of methodological standards for anthropomorphized misalignment research (AMR). The paper considers the concept of interpreting LLM behaviors (e.g., deception, emergent misalignment, shutdown resistance) and argues that current studies often over-interpret or mis/over-claim behavioral results, due to conceptual ambiguity, small/fragile datasets, evaluation sensitivity (e.g., LLM judges), and insufficient causal interventions. The main contribution is a three-level evidence framework (L1 behavioral, L2 functional downstream effects, L3 causal-mechanistic attribution) plus a checklist and recommendations for more calibrated claims.

**Position:**

Yes

**Position In Title:**

Yes

**Related Work:**

3

**Strengths And Weaknesses:**

### Strengths

* Timely and relevant: improves rigor/communication in an important and fast-moving safety area.
* Useful framework: L1/L2/L3 provides a concrete lens for matching claims to evidence. (although I would argue for scientific claims to be true in L2, people should provide L3 evidence)
* Practical guidance: recommendations (ablations, distributional diversity, alternative-explanation tests, interventionist evidence) are actionable. (although these are still very high-level and abstract)
* Good examples: highlights judge sensitivity, probe false positives, and capability degradation as confounders.

### Weaknesses

* The L2 vs L3 distinction can feel overstated for ML audiences: robust functional effects often already imply strong constraints on internal structure; the paper could clarify what *additional* evidence differentiates L3 beyond “mechanism exists” vs “mechanism identified.”
* Some critiques are primarily methodological/illustrative rather than decisive; clearer criteria for “sufficient” evidence would help. And more concrete details are encouraged.
* The precaution vs rigor trade-off is acknowledged but could be developed more (safety decisions may rationally act on weaker evidence).

**Support:**

2

---

> ### Author Rebuttal · Authors · 2026-03-31
>
> We thank the reviewer for the thoughtful assessment and constructive questions. We are glad that you find the L1/L2/L3 framework useful, and the examples and recommendations actionable. We appreciate the push for more decisive criteria.
>
> ### On added values for the community beyond raising the problem statement:
> We appreciate this concern. While many methodological pitfalls are discussed informally, they are **not yet codified as shared standards in AMR and are not widely implemented in assessing or communicating the latest research outputs**. Our contribution is to:
> - Systematize failure modes along a shared pipeline (S1 to S4).
> - Introduce an evidence level framework (L1-L3) to align claim strength with what experiments establish.
> - Provide operational recommendations and a diagnostic checklist (Section 4.2, Appendix B).
> - Provide concrete demonstrations of key failure modes (Appendix C).
>
> While Section 2 outlines the differences between our work and prior critiques, we are happy to expand this discussion if you identify any other specific works that would further contextualize our contribution.
>
> ### Q1. Operationalizing L3 evidence
>
> Convincing L3 evidence requires demonstrating through targeted intervention that a specific internal factor causally mediates the behavior, and that the effect is specific rather than general degradation (Section 4.1). What this looks like depends on the behavior: for sycophancy, it could mean showing that a modified RLHF algorithm eliminates sycophantic outputs, or that ablating a specific circuit reduces sycophancy while leaving other capabilities intact. The shared requirement is intervention with specificity, as stated in R10. In our Checklist (Appendix B, S4), we already listed many of these requirements implicitly. We will edit the appendix to explicitly label them by evidence level.
>
>
> ### Q2. Deployment Framing
>
> We would like to clarify that our primary goal is to establish scientific standards for calibrating claims to evidence, rather than to set policy thresholds. Translating evidence levels into specific regulatory recommendations involves additional considerations (risk tolerance, use-case context, stakeholder interests) and risks conflating 'what has been established?' with 'what should we do about it?'. This is important follow-up work, but it falls outside the scope of this paper.
>
> ### W1. Distinguishing L2 vs L3
> We appreciate this point and agree that the distinction can be made more clearly. If a paper claims to have mitigated LLM-induced misconceptions, it must: (1) First establish that a model induces misconceptions in humans (this is strong L2); (2) Run an intervention and measure changes in misconceptions; (3) Provide evidence that the intervention does not target correlated behaviors, but indeed the misconception-inducing behavior itself. (2) and (3) correspond to strong L3 evidence.
>
> L3 evidence is required whenever a paper claims to have identified or mitigated a causal factor for a behavior. Appropriate L1 or L2 evidence is a prerequisite to establishing a strong L3.
> In our revised version, we will clarify this distinction and give brief, concrete examples to make the boundary clearer.
>
> ### W2. Criteria for "sufficient" evidence
>
> We appreciate the reviewer raising this distinction. Our Checklist focuses on necessary evidence — the minimum empirical support without which a claim cannot be considered substantiated. We deliberately avoid prescribing a universal sufficiency threshold, as what constitutes sufficient evidence depends on the specific claim, threat model, and deployment context. However, we agree that making this scope explicit strengthens the paper. We will clarify that our Checklist targets necessary (not sufficient) conditions, and we will add some concrete examples to illustrate the boundary more tangibly.
>
> ### W3. Precaution vs Rigor
>
> We appreciate this point and agree that early, imperfect evidence can justify further investigation and provisional safeguards, especially when risks are asymmetric. Our position is that precaution and rigor are complementary, not opposed. We do not say 'wait for L3 before acting': L1 can justify monitoring and continued investigation, and a strong L2 can justify deployment restrictions. Evidence-based claims can benefit efficient resource allocation and decision-making, even in the early stages.  We will explicitly address these nuances in our revised version.
>
> ### Ethics Flag
> To the best of our knowledge, this paper does not violate any research ethics as outlined in the [ICML standards for ethical conduct of research](https://icml.cc/Conferences/2026/ResearchEthics).
>
> We thank you again for the constructive feedback. We believe the suggested clarifications will strengthen the paper, and we welcome further discussions.

---

> > ### Author Rebuttal · Reviewer_uKr5 · 2026-03-31
> >
> > Thank you for the rebuttal and clarifications. I now better understand the intended role of the paper as a position piece.
> >
> > That said, my main concern remains about contribution rather than correctness of the overall opinion. I agree that many of the cautions raised are sensible, and I also see why other reviewers may value the paper as a field-organizing effort. However, I am still not fully convinced that the paper adds enough beyond systematizing concerns that are already informally present in the community.
> >
> > In particular, while the S1-S4 pipeline, the L1/L2/L3 framing, and the checklist are useful, I still find the contribution somewhat high-level. The rebuttal improves clarity, but I do not yet see a sufficiently sharp operational standard or enough additional substance to make this feel like a strong standalone contribution, as opposed to a useful synthesis of existing concerns.
> >
> > So my concern is less that I disagree with the paper’s message, and more that I remain uncertain about whether the paper contributes enough, in itself, for acceptance at the current level.

---

### Official Review · Reviewer_2VnY · 2026-03-13

**Significance:** 3
**Argument Clarity:** 3
**Rating:** 5
**Confidence:** 3

**Questions:**

N/A.

**Alternative Views Section:**

Yes

**Compliance With Llm Reviewing Policy A Conservative:**

Affirmed.

**Discussion Potential:**

3

**Paper Summary:**

This paper advocates that anthropomorphic misalignment research (AMR), which is a family of alignmentoriented studies that investigate safety-relevant failure modes in AI described through human-like characteristics, etc. Studying these failure modes are inherently challenging because (1) concepts that relate to human-like characteristics are hard to precisely define and measure, (2) the dataset constructed for the concepts are small and can struggle to accurately carry the desired concepts, and (3) the experiments conducted using the dataset lack ablation studies and proper de-confounding. Therefore, the authors suggest a framework that allow for collecting evidence at behavior, function, and causation level through experiments. Recommended practices include better concept declearation, scaling up dataset, measuring interventional robustness.

**Position:**

Yes

**Position In Title:**

Yes

**Related Work:**

3

**Strengths And Weaknesses:**

**Strength**
1. The discussion on the limitation of current anthropomorphic misalignment research is sufficient. Multiple limitations are supported by case studies and quantitive results. Therefore, the motivation of this paper is solid.
2. The topic of anthropomorphic misalignment research is very relevant to the ICML sommunity and likely to insipre discussions and improved approaches.
3. The argument is clear. This paper recommends three types of evidences and outline approaches to collect them.

**Weakness**
1. The position "Needs Stronger Evidence" is a bit vague. It would be helpful if the position can better explain what types of evidence are strong. Having a clear definition in the position also makes the discussion of alternative view more informative.
2. Some of the advocates overlap with dataset research paper, such as sufficient scale and distributional diversity. Maybe seperating the dataset measurement and how to better use a dataset can make the advocates more clear.

**Support:**

3

---

> ### Author Rebuttal · Authors · 2026-03-31
>
> We thank the reviewer for the positive assessment and thoughtful feedback. We are encouraged that you find the motivation solid, the topic relevant to the community, and the arguments clear.
>
> > The position "Needs Stronger Evidence" is a bit vague. It would be helpful if the position can better explain what types of evidence are strong. Having a clear definition in the position also makes the discussion of alternative view more informative.
>
> We appreciate this feedback and agree that a clear definition of "strong evidence" is important. We intended Section 4.1 to operationalize this, but we can surface it earlier in the paper and make it more explicit.
>
> In our paper, “strong evidence” is not a single absolute bar. It is claim-relative, and it means (i) using the appropriate evidence level for the claim, and (ii) meeting the corresponding methodological requirements for that level. Concretely:
> - L1 (behavioral): establishes that a behavior occurs under a given setup. Claims about the existence of a behavior can be supported here.
> - L2 (functional): shows that the behavior produces consistent safety-relevant downstream effects across variations. Claims about reliable real-world impact require this level.
> - L3 (causal-mechanistic): identifies internal components that causally generate the behavior via interventions and alternative explanation testing. Claims about mechanism, intent, or internal goals require this level.
>
> Under this framework, "stronger evidence" means closing this gap by matching the claim to the appropriate evidence level.
>
> In our revised version, we will extend the position statement in the introduction to mention these levels of evidence and link to section 4.1. We would appreciate your feedback on whether this addresses your concern.
>
> > Some of the advocates overlap with dataset research paper, such as sufficient scale and distributional diversity. Maybe seperating the dataset measurement and how to better use a dataset can make the advocates more clear.
>
> This is an insightful distinction, and we agree that the current presentation can blur the distinction between “dataset construction” and “how the dataset is used in AMR.” We do not intend to claim novelty over the broader dataset methodology literature. Rather, we include R4 and R5 because small, homogeneous, or confounded AMR datasets directly drive the kinds of overinterpretation we highlight (e.g., surface-cue artifacts and concept-to-dataset slippage).
>
> To make the recommendations clearer, we will revise the write-up to more explicitly separate:
> - Dataset properties (R4, R5): scale, distributional diversity, and surface-feature controls during construction and operationalization.
> - Evaluation discipline that uses the dataset (already in R6 to R9 and Appendix B): stress tests and negative controls, paraphrase and setting ablations, sensitivity reporting, and tests of alternative explanations.
>
> Concretely, we will (i) tighten the wording of R4 and R5 to stay strictly about construction, and (ii) add clearer cross-references from R8 and R9 (and the checklist) to the “how to use a dataset” practices you list, so the distinction is explicit to the reader. We would welcome further clarification if this does not fully capture your suggestion, but this separation aligns with our understanding of your concern.

---

> > ### Author Rebuttal · Reviewer_2VnY · 2026-04-06
> >
> > My concerns have been adequately addressed and I keep my rating - 5: Accept.

---

### Official Review · Reviewer_qmyT · 2026-03-13

**Significance:** 4
**Argument Clarity:** 4
**Rating:** 6
**Confidence:** 4

**Questions:**

n/a

**Alternative Views Section:**

Yes

**Compliance With Llm Reviewing Policy A Conservative:**

Affirmed.

**Discussion Potential:**

4

**Final Justification:**

See rebuttal ack.

**Paper Summary:**

This paper studies the subfield of "anthropomorphic misalignment research" (AMR), which the authors define as alignment-related research focused on concerns where the 'behavior' of AI system resembles anthropomorphic qualities (e.g., deception, scheming, etc; see S2), and argues that many works in this field "need stronger evidence." The authors take a scientific/experiment design perspective on the different failure modes that may arise in each of four stages of an AMR study (S3); in addition to citing examples from the AMR literature, the authors also run a set of small experiments as further evidence of their claims. The authors also provide concrete recommendations and examples of best practice (S4) and address common justifications for the widespread proliferation of [questionable-quality] AMR (S5).

**Position:**

Yes

**Position In Title:**

Yes

**Related Work:**

4

**Strengths And Weaknesses:**

While the stated position is relatively mild --- that "many" AMR papers need more evidence to claim the existence of such misalignment --- the paper itself is a very strong case for that position (in fact, I think it would be reasonable to have supported a much stronger stated position, though I understand the rhetorical benefits of a more subdued headline in this case). I appreciated that the authors made citations to examples of AMR work that exhibited the identified limitations, but also that additional experiments were run. I also found the definition of AMR as a field to be itself a valuable contribution, as a way to categorize different types of misalignment research and reason about type-specific failure modes.

Something that I think would strengthen the paper is a survey of what is currently known about AMR, and how that changes when stricter evidentiary standards are applied. For example, something along the lines of "The literature in AMR would suggest that X, Y, and Z are true/possible/etc (cites). However, applying our lens to these papers substantially changes how one should interpret what is actually currently true/possible/etc: { insert findings }." It would also be useful (as a reader) to see you pull a set of papers out into a table and give a high-level summary of what claims are actually supported by the evidence in each paper, vs what the paper appears to be claiming. (That said, I also understand why the authors might want to avoid calling out specific papers in more detail!)

**Support:**

4

---

> ### Author Rebuttal · Authors · 2026-03-31
>
> We thank the reviewer for the very positive and thoughtful evaluation. We are especially encouraged that you find the argument strong, the framing of AMR useful, and the combination of analysis and experiments compelling. We address your suggestions below.
>
> > ...it would be reasonable to have supported a much stronger stated position, though I understand the rhetorical benefits of a more subdued headline…
>
> We appreciate this perspective and are glad the argument comes across as strong. While we agree that the evidence presented could support a more provocative headline, we deliberately chose a measured title. Our goal is to provide a constructive path forward that invites the AMR community to adopt higher standards without dismissing the underlying importance of the research. A stronger headline risks being interpreted as dismissing the entire line of work, which could be counterproductive given that we view many of these questions as important but currently under-supported. We will consider minor refinements to improve clarity, but we prefer to retain a framing that emphasizes calibration rather than rejection.
>
> > ...a survey of what is currently known about AMR, and how that changes when stricter evidentiary standards are applied.
>
> This is an excellent point. Our paper already does this implicitly throughout Section 3, but we agree that making it explicit would strengthen the presentation. When our framework is applied, several high-profile findings shift in interpretation:
>
> - Deception: Current literature often suggests that models possess strategic intent when producing deceptive-looking answers. However, applying R9 shows that these instances are merely reflexive responses to cues or role-play artifacts. [1]
> - Shutdown Resistance: What is often reported as "self-preservation" appears, under stricter scrutiny, to be a byproduct of instruction ambiguity. [2,3]
> - Emergent Misalignment (EM): While often attributed to specialized "internal personas", the literature suggests these may instead be a generic erosion of safety guardrails due to OOD shifts. [4,5,6]
>
> We will integrate this "before and after" perspective more visibly in the revision.
>
> > ...pull a set of papers out into a table and give a high-level summary of what claims are actually supported... vs what the paper appears to be claiming.
>
> We considered this during drafting. We opted against it for two reasons. First, a faithful representation would require substantial context per paper to avoid mischaracterization, which is difficult within space limits. Second, and more importantly, we want the framework to function as a constructive tool that encourages adoption rather than a scorecard that singles out individual contributions. Since AMR is still pre-paradigmatic, with definitions and evaluation standards not yet established, a paper-by-paper audit risks devolving into disputes over individual cases rather than advancing shared standards. We believe the representative examples throughout the text, combined with the pipeline-level failure modes and the "before and after" perspective above, achieve this at the right level of abstraction.
>
> We thank you again for the constructive feedback.
>
> **Sources:** [1] Smith et al., "Difficulties with Evaluating a Deception Detector for AIs," 2025. [2] Rajamanoharan & Nanda, "Self-Preservation or Instruction Ambiguity? Examining the Causes of Shutdown Resistance," 2025. [3] Schlatter et al., "Shutdown Resistance in Large Language Models," 2025. [4] Giordani, "Re-Emergent Misalignment: How Narrow Fine-Tuning Erodes Safety Alignment in LLMs," 2025. [5] Woodruff, "Aesthetic Preferences Can Cause Emergent Misalignment," 2025. [6] Wang et al., "Persona Features Control Emergent Misalignment," 2025.

---

> > ### Author Rebuttal · Reviewer_qmyT · 2026-04-03
> >
> > I appreciate the decisions you made and find them reasonable choices - looking forward to seeing the final version of the paper!

---

### Official Review · Reviewer_d7b6 · 2026-03-17

**Significance:** 4
**Argument Clarity:** 4
**Rating:** 6
**Confidence:** 4

**Questions:**

(no questions)

**Alternative Views Section:**

Yes

**Compliance With Llm Reviewing Policy A Conservative:**

Affirmed.

**Discussion Potential:**

4

**Final Justification:**

The other reviews and the authors rebuttals confirmed my original review. This position paper is a thoughtful and measured contribution to the literature on AMR, and it should be accepted.

**Paper Summary:**

The paper describes the methodological challenges for research in anthropomorphic misalignment research (AMR). The paper characterizes the pipeline typically employed in such research, describes specific methodological challenges in each stage of that pipeline, and then suggests approaches to mitigating those challenges.

**Position:**

Yes

**Position In Title:**

Yes

**Related Work:**

4

**Strengths And Weaknesses:**

The authors wisely frame entire discussion as one of uncertainty: Because of the methodological problems in much of the existing AMR research, we don’t know whether specific types of apparent behavior are really causes for concern.

The shared AMR pipeline outlined in Section 2.1 is both a useful framework for the paper and a contribution unto itself. While seemingly obvious, it helps identify specific instances of potential problems.

The paper has extensive and wide-ranging references that should make it useful merely as a resource on methodological critique of AMR.

The authors report the results of several experiments in small boxes throughout the paper. The full results are provided in the appendices. The experimental results are highly relevant and helpful, and the fact that they are reported in boxes (with details in the appendix) prevents the flow of the paper from being interrupted. Great work!

At least to this reader, the context and overall relevance of Experiment 3 is unclear. The paper would be improved by providing more context and revising the description for readers unfamiliar with that context.

The authors provide somewhat concrete actions to take to improve AMR validity, while acknowledging that some issues "…are rooted in conceptual or epistemic limits that demand new theoretical insight from outside the immediate AMR community." In particular, they point to studies that have done particularly well dealing with certain issues.

In some cases, avoiding or mitigating the issues that the authors identify may require a wholesale change in the basic mindset of researchers. Recommending adoption of specific methodologies assumes that the primary problem is merely one of awareness and knowledge among researchers, whereas many of the existing problems may be partially or predominantly rooted in the motivations of some researchers to publish as many papers as possible as quickly as possible and to minimally clear any existing methodological bar. That said, this paper may motivate reviewers of those papers to demand better methodology, regardless of the motivations of the authors of those papers.

The “Alternative Views” section is reasonable and covers the alternatives that I would have expected (and which are often put forward in the community).

**Support:**

4

---

> ### Author Rebuttal · Authors · 2026-03-31
>
> We thank the reviewer for the very positive and thoughtful evaluation. We are especially encouraged that you find the framing around uncertainty appropriate, the AMR pipeline useful, and the experimental presentation effective. Below we address your concerns.
>
> > At least to this reader, the context and overall relevance of Experiment 3 is unclear.
>
> We appreciate this feedback and will clarify the relevance of Experiment 3 in the final version. The core intent of this experiment is to demonstrate that correlational evidence, the current standard for "detecting" deception in models, is often an artifact of surface-level features rather than the model's actual intent. By showing that these probes "fire" in harmless, honest scenarios like sarcasm or recital (where a model is simply following instructions), we provide empirical support for challenge C8 (spurious correlations).
> We will revise the description to clearly state this goal upfront, and add a concluding sentence explicitly linking the result back to the broader methodological concern.
>
> > ...existing problems may be partially or predominantly rooted in the motivations of some researchers to publish... That said, this paper may motivate reviewers of those papers to demand better methodology…
>
> We agree that this is an important point. We acknowledge that while our recommendations address technical awareness, they do not directly address the field's underlying incentive structures. However, as you suggest, we believe that by establishing these clear evidence levels (L1-L3) and a diagnostic checklist, we provide the community (and specifically reviewers) with the objective tools needed to demand higher standards. We hope this paper acts as a catalyst for a "bottom-up" shift in what the community considers acceptable evidence for high-stakes safety claims.

---

> > ### Author Rebuttal · Reviewer_d7b6 · 2026-04-05
> >
> > Given my highly positive review, there was little for the authors to address, though they successfully addressed the few remaining critical points I raised.

---

### Decision · Program_Chairs · 2026-04-30

**Decision:**

Accept (spotlight)

**Comment:**

The decision is to accept that paper.

The paper provides a constructive critique of anthropomorphic misalignment research that systematizes both the research area and the sets of pitfalls that can arise in current research workflows. Reviewers recognized the value of both pieces, as well as the actionable recommendations for conducting and evaluating AMR. The authors are very careful about exactly what role they hope their recommendations should play (e.g., necessary but not sufficient conditions, assessing "what is happening" vs "what should we do about it", etc). In a growing research area that is "pre-paradigmatic", this sort of position could have major impact on the direction of the field.